# Simulation of an AlGaInAs/InP Electro-Absorption Modulator Monolithically Integrated with Sidewall Grating Distributed Feedback Laser by Quantum Well Intermixing

Xiao Sun [1,*], Weiqing Cheng [1], Yiming Sun [1], Shengwei Ye [1], Ali Al-Moathin [1], Yongguang Huang [2], Ruikang Zhang [2], Song Liang [2], Bocang Qiu [3], Jichuan Xiong [4], Xuefeng Liu [4], John H. Marsh [1] and Lianping Hou [1]

1   James Watt School of Engineering, University of Glasgow, Glasgow G12 8QQ, UK
2   Institute of Semiconductors, Chinese Academy of Sciences, No. A35, East Qinghua Road, Haidian District, Beijing 100083, China
3   Institute of Atomic and Molecular Science, Shaanxi University of Science and Technology, Xi'an 712081, China
4   School of Electronic and Optical Engineering, Nanjing University of Science and Technology, Nanjing 210094, China
*   Correspondence: x.sun.2@research.gla.ac.uk

**Abstract:** A novel AlGaInAs/InP electro-absorption modulated laser (EML) with a simple fabrication process is proposed, in which the electro-absorption modulator (EAM) has a 10 nm blueshift induced by quantum well intermixing (QWI) and is monolithically integrated with a sidewall grating distributed-feedback (DFB) laser working at 1.55 μm wavelength. The extent of the QWI process is characterized by a diffusion length. The quantum confined Stark effect (QCSE) is simulated in terms of extinction ratio (ER) and chirp for bias electric fields from 0 kV/cm to 200 kV/cm and for different amounts of intermixing. The results indicate that for a 150 μm-long EAM with a 10 nm blueshift induced by QWI, an ER of 40 dB is obtained at 2.5 V reverse bias with no penalty in chirp compared to an as-grown quantum well (QW) and the insertion loss at 0 V bias is 0.11 dB for 1.55 μm operation wavelength. The simulated –3 dB bandwidth of the electrical to optical power response is 22 GHz.

**Keywords:** AlGaInAs; multi-quantum well (MQW); sidewall grating; quantum well intermixing (QWI)



## 1. Introduction

Electro-absorption modulators (EAMs) which operate using the quantum-confined Stark effect (QCSE) are attractive due to their low power consumption, small size, large bandwidth and potential for monolithic integration with other components [1,2]. Several electro-absorption modulated lasers (EMLs), composed of EAMs monolithically integrated with distributed feedback (DFB) lasers, have been investigated in the InGaAsP material system [3–6] and AlGaAs system [7]. EMLs based on quantum dots (QDs) have been reported in [8,9] and achieve a 10 dB extinction ratio (ER). Compared with QD structures, multi-quantum well (MQW) structures have a larger optical confinement factor, which is proposed to realize a higher ER. MQWs also have a larger carrier density by two orders of magnitude and a better QCSE effect than QDs [10]. For EMLs to be used in very short-reach (VSR) systems and passive optical networks (PONs), it is highly desirable to realize uncooled operation and simplify the device structure and fabrication processes to minimize production costs while delivering devices with good reliability. Compared with InGaAsP material, EMLs based on AlGaInAs have a larger conduction band discontinuity and smaller valence band discontinuity, which are more suitable for uncooled device operation. AlGaInAs EMLs have previously been proposed using a relatively complicated and time-consuming butt-joint regrowth technique [3], where conventional buried grating DFB lasers

and regrowth EAMs were used. To simplify the fabrication process, described in [11–13], sidewall grating (SWG) DFB lasers and identical epitaxial layer (IEL) EAMs were used. Because the EAMs used an identical epitaxial layer to the DFB laser, the Bragg wavelength had to be set at 1.56 μm to reduce the absorption loss in the EAM section for a DFB laser with a photoluminescence (PL) wavelength at 1.53 μm. In [4], EMLs were reported based on the quantum well intermixing (QWI) technique, but the DFB lasers used conventional buried gratings, which still need at least two steps of the Metal-Organic Vapor Phase Epitaxy (MOVPE) processing. Their maximum static ER was only 14 dB at 4 V EAM reverse bias. However, using EMLs based on QWI can lead to reduced threshold currents and insertion losses and also to an enhanced modulation performance of the light source [14].

To simplify the fabrication process and further reduce the insertion loss of the EAM while obtaining a relatively high ER value at low EAM reverse voltage, a novel integrated EML is proposed using an SWG DFB laser and QWI EAM. Compared with conventional DFB lasers with buried gratings, which use complicated fabrication technologies, including etch and regrowth steps to complete the epitaxy of the laser structure after grating definition, DFB lasers based on SWGs have several advantages, such as regrowth-free fabrication processes, increased design flexibility and readily allowing the use of Al-containing epitaxial structures [15]. The gratings may be defined simultaneously with the ridge waveguide, significantly simplifying the device manufacturing process. Compared with conventional selective etch and regrowth techniques for photonic integration, the technique of post-growth processing based on QWI offers a simple, flexible and low-cost alternative. When an electric field bias is applied to the EAM's quantum wells (QWs), the edge of the absorption spectrum is shifted to lower energies for both heavy hole (HH) and light hole (LH) transitions, an effect known as the QCSE. The QCSE has been intensely investigated for EAMs based on InGaAs/InP [16], InGaAs/AlGaAs [17] and InGaAsP/InP [18], but has not been reported for 1.55 μm AlGaInAs/InP EAMs. Here, we establish the QWI model, calculating the profiles of the wavefunctions in the conduction and valence bands, the absorption spectrum, ER and chirp factor for an as-grown AlGaInAs/InP EAM and an EAM with QWI. For the intermixed QW structure, the absorption spectra show larger movements of the exciton peak and larger Stark shifts than for the as-grown QW, which is promising for the design of EAMs operating at high-speed and low drive voltage. For a 150 μm-long EAM with a 10 nm blueshift, an ER of –40 dB is predicted at 2.5 V EAM reverse bias.

## 2. The Model Theory

The EAM device is based on a commercially available 1550 nm AlGaInAs/InP LD structure [19]. This wafer contains five 6 nm-thick compressively strained (+1.2%) AlGaInAs QWs and six 10 nm-thick tensile strained (–0.3%) AlGaInAs quantum barriers (QBs). The MQW layer is sandwiched by two 60 nm-thick graded-index separate confinement heterostructure (GRINSCH) AlGaInAs layers, whose Al compositions are graded from 0.42 to 0.34 (0.34 is at the QB side) and two 60 nm-thick AlGaInAs waveguide layers. MOVPE is used to grow this structure on a sulfur-doped InP substrate. According to quantum mechanical selection rules, the compressive strain in the AlGaInAs QWs results in TE-polarized gain.

The extent of the QWI process is characterized by a diffusion length $L_D$. To calculate the optical properties of intermixed material, Fick's second law of diffusion is used to model the QW profile [20] and its PL spectrum during intermixing. In the model, the degree of intermixing $N (z, L_D)$ is represented by the diffusion length $L_D$ on the group III substance, as shown in Equation (1):

$$N(Z, L_D) = (N_W - N_B)\left[1 - \frac{1}{2}\text{erf}\left(\frac{z - \frac{L_W}{2}}{2L_D}\right) + \frac{1}{2}\text{erf}\left(\frac{z + \frac{L_W}{2}}{2L_D}\right)\right] \qquad (1)$$

where $N_W$ and $N_B$ are the initial atomic mole fractions for QW and QB materials, respectively; $z$ is the quantization direction along the growth axis (QW centered at $z = 0$); "erf" denotes the error function; and $L_W$ is the QW width. The experimental and calculated bandgap shifts can be fitted by an appropriate choice of $L_D$. Figure 1 shows the calculated Al and Ga profiles for the as-grown structure and intermixed material where $L_D$ is 0.45 nm. The bandgap shifts of the conduction and valence band, which are due to the strain in the well and barrier, are included in the calculation. The values of the material parameters can be found in Table 1, where $\gamma_1$, $\gamma_2$ and $\gamma_3$ are Luttinger parameters of the valence band [21].

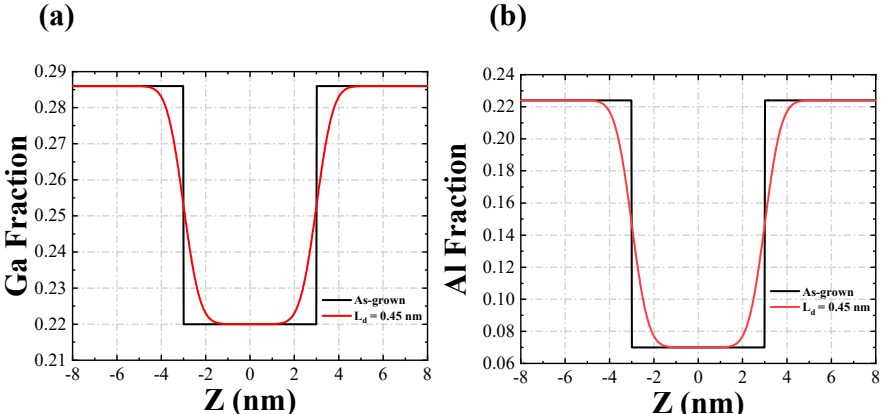

**Figure 1.** (**a**) Calculated Ga and (**b**) Al fraction profiles for the as-grown structure and intermixed material, where $L_D$ is 0.45 nm.

**Table 1.** The AlGaInAs material parameters [22–25].

| Parameters | Symbol (Unit) | $Al_xGa_yIn_{1-x-y}As$ |
|---|---|---|
| Lattice constant | $a$ (Å) | $5.6x + 5.6533y + 6.0584(1 - x - y)$ |
| Elastic stiffness constant | $C_{11}$ ($10^{11}$ dyn/cm$^2$) | $1.25x + 1.1879y + 0.8329(1 - x - y)$ |
| | $C_{12}$ ($10^{11}$ dyn/cm$^2$) | $0.534x + 0.5376y + 0.4526(1 - x - y)$ |
| Hydrostatic deformation potential for conduction band | $ac$ (eV) | $-5.64x - 7.17y - 5.08(1 - x - y)$ |
| For valence band | $a_v$ (eV) | $2.47x + 1.16y + 0.66(1 - x - y)$ |
| Shear deformation potential | $b$ (eV) | $-1.5x - 1.7y - 1.8(1 - x - y)$ |
| Electron effective mass | $m_e/m_0$ | $0.15x + 0.067y + 0.023(1 - x - y)$ |
| Heavy-hole effective mass | $m_{hh}/m_0$ | $0.7x + 0.5y + 0.517(1 - x - y)$ |
| Light-hole effective mass | $m_{lh}/m_0$ | $0.1415x + 0.088y + 0.024(1 - x - y)$ |
| Valence band parameter | $\gamma_1$ | $3.45x + 6.8y + 20.4(1 - x - y)$ |
| | $\gamma_2$ | $0.68x + 1.9y + 8.3(1 - x - y)$ |
| | $\gamma_3$ | $1.29x + 2.73y + 9.1(1 - x - y)$ |

The QW conduction and valence discrete sub-band edges can be calculated from the effective mass Schrödinger equation as [26]:

$$-\frac{\hbar}{2m_r(z)}\nabla^2\chi_r + U_r(z)\chi_r = E_r\chi_r \tag{2}$$

where $\hbar$ is Planck's constant divided by $2\pi$; $\chi$ is the envelope wave function; $E$ is the electron energy level; $m_r$ is the effective mass in the vertical direction; and $U_r$ is the energy band potential. The subscript $r$ corresponds to the conduction band and the HH and LH valence bands. This equation can be solved by the finite difference method to obtain the $i$th conduction and $j$th valence sub-band energies $E_{ci}$ and $E_{vj}$. The energies of the interband transitions and the conduction bandgap offset of the *QW* are:

$$E_{ij}^{QW} = E_g^{strain} + E_{ci} + E_{vj} \tag{3}$$

$$\Delta E_c = Q\Delta E_g \tag{4}$$

where $Q$ is the band offset ratio; for AlGaInAs QWs and QBs, $Q$ is 0.72 and the strained QW bandgap energy is $E_g^{QW} = E_{11}^{QW}$. The overlap integral is presented as:

$$\langle \chi_{ci} | \chi_{vi} \rangle = \int_{-z_b}^{z_b} \chi_{ci}(z)\chi_{vi}^*(z)dz \tag{5}$$

where $z_b$ *is* taken to be the boundary where $\chi_{ci}$ and $\chi_{vi}$ tend to 0. The corresponding exciton resonance absorption spectra for transitions between the different conduction and valence band discrete energy levels ($\alpha_{ex}$) and the absorption spectra between the conduction and valence band transition ($\alpha_{con}$) are expressed as [16]:

$$\alpha_{ex} = \sum_{i,j} \frac{4e^2\hbar^2\omega p_{cv}M_0}{\varepsilon_0 cnm_0^2 E_{ex(i,j)}^2 L_W} |\langle \chi_{ci} | \chi_{vi} \rangle|^2 B\left(\hbar\omega - E_{ex(i,j)}\right) \tag{6}$$

$$\alpha_{con} = \frac{2e^2\mu^*\omega p_{cv}^2 M_0}{\varepsilon_0 cnm_0^2 E_{cv}^2 L_W} \sum_{i,j} |\langle \chi_{ci} | \chi_{vi} \rangle|^2 \int_{E_{cv(i,j)}}^{\infty} S\left(E, E_{cv(i,j)}\right) L(E, \hbar\omega)dE \tag{7}$$

The total absorption is:

$$\alpha = \alpha_{ex} + \alpha_{con} \tag{8}$$

where $n$ is the refractive index; $c$ is the velocity of light in vacuum; $\varepsilon_0$ is the permittivity of vacuum; $\mu^*$ is the reduced mass in the transverse direction; and $L_W$ is the width of the as-grown QW. The Sommerfeld enhancement factor ($S$) and the Lorentzian broadening factor ($L$) average matrix element for the Bloch state ($M_0$) are:

$$M_0 = \frac{m_0^2 E_g (E_g + \Delta_0)}{12 m_e^* \left(E_g + \frac{3}{2}\Delta_0\right)} \tag{9}$$

$$S\left(E, E_{cv(i,j)}\right) = \frac{2}{1 + \exp\left(-2\pi\frac{\sqrt{E - E_{cv(i,j)}}}{R_y}\right)} \tag{10}$$

$$L(E, \hbar\omega) = \frac{\Gamma}{\pi\left[(\hbar\omega - E)^2 + \Gamma^2\right]} \tag{11}$$

where $\Delta_0$ is the spin–orbit splitting and $R_y = \mu^* e^4 / 32\pi^2\varepsilon^2 h^2$ is the Rydberg energy; $E$ is the photon energy; $\Gamma$ is the bound state linewidth broadening factor (half-width half maximum); $E_{ex}$ is the exciton transition energy; and $p_{cv}$ is the polarization factor which is different for TE and TM modes. When the polarization vector is parallel to the quantum-well layers, the polarization factor is expressed as $p_{cv}^{TE}$ = 1.5 (HH), 0.5 (LH), $p_{cv}^{TM}$ = 0 (HH), 2 (LH) [16]. The chirp factor $\alpha_H$ of the EAM is calculated from:

$$\alpha_H = \frac{\delta n}{\delta k} = \frac{4\pi}{\lambda}\frac{\delta n}{\delta \alpha} \tag{12}$$

where $\delta\alpha$ is the differentiation of the absorption coefficient to photonics energy and $\delta n$ is the change in refractive index, which can be calculated using the Kramers−Krönig relationship [27]:

$$\delta n(E) = \frac{c\hbar}{\pi}\int_0^{\infty} \frac{\delta\alpha(E)}{E'^2 - E^2}dE' \tag{13}$$

## 3. Simulation and Results

The structure of the EML device is shown in Figure 2a. The SWG DFB is 600 μm long and the EAM is 150 μm long. Based on our former universal damage-enhanced QWI experimental results reported in [28], the QWI transition length is around 30–50 μm.

The EAM and DFB in this design are separated by a 50 μm-long isolation region. The DFB laser and EAM were shallow- and deep-etched, respectively, with the same ridge waveguide width of 2.5 μm. The calculated intensity reflectivity at the interface between the shallow-etched DFB section and the deep-etched EAM section is about 0.8%, which may have negative effects on the wavelength stability, side-mode suppression ratio (SMSR) and output waveform of the device. To overcome this performance degradation, partially corrugated gratings for DFB lasers can be used which have high immunity to residual reflection from the interface or the DFB and EAM facets [29,30]. The DFB laser has a first-order SWG with a 0.6 nm recess depth and a 243 nm grating period with a 50% duty cycle, as shown in Figure 2a. A quarter wavelength shift section was inserted at the center of the DFB laser cavity to ensure single longitudinal mode oscillation. The PL wavelength of the EML at room temperature is 1531 nm. The device fabrication and the universal damage-enhanced QWI process have been reported previously [31]. Figure 2b presents the relationship between the electric field inside the MQW layer and the applied biased voltage ($V_b$). It can be seen that the built-in electric field is 59 kV/cm at 0 V bias. Figure 2c presents the band diagram of the as-grown sidewall EML structure.

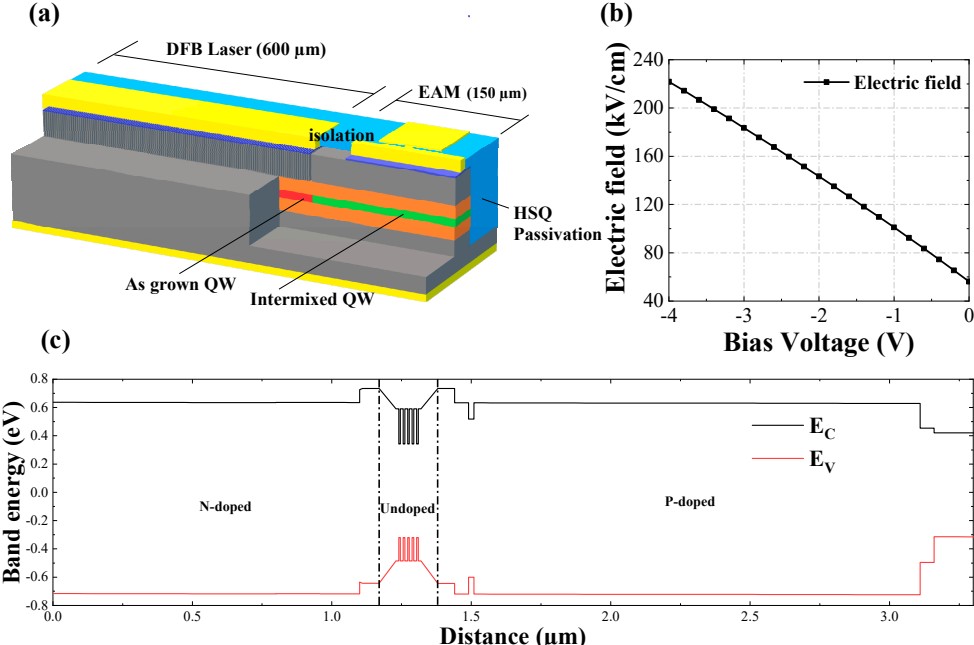

**Figure 2.** (**a**) Schematic structure of the EML, (**b**) simulated electric field inside the MQW layer as a function of applied biased voltage $V_b$ and (**c**) band diagram of the as-grown epitaxial structure.

Figure 3 shows the effect of the electric field on the electron and HH ground state wave functions for $L_D = 0$ nm and $L_D = 0.45$ nm. The field-induced shift of the electron eigenfunction is to the right and the HH eigenfunction to the left. Thus, in the presence of an electric field, the overlap integral between the electron and HH wave functions is reduced.

Figure 4a,b present the TE polarized absorption coefficient spectrum for the as-grown QW and diffused QW $L_D = 0.45$ nm structure, with an electric field bias from 0 to 200 kV/cm in steps of 50 kV/cm, calculated from Equation (8). The investigation found that for the as-grown QW, the wavelength of the exciton peak is 1531 nm, and the absorption coefficient at 1.55 μm for 0 kV/cm is 400 /cm. After QWI with a diffusion length of $L_D = 0.45$ nm, there is a 10 nm blueshift in the exciton peak and the absorption coefficient is reduced to 2 /cm at 1.55 μm wavelength for 0 kV/cm. Figure 4c shows the wavelength shift of the exciton peak as a function of $L_D$: as $L_D$ is increased, the wavelength blueshift increases monotonically. Figure 4d shows the square of the overlap integral of the wavefunctions between the first conduction and the HH valence sub-bands as a function of electric field

for as-grown and intermixed QW with $L_D$ = 0.45 nm, respectively. The latter square of integral overlap is slightly lower than the former one when the electric field is less than 75 kV/cm, but the difference is negligible when the electric field is more than 75 kV/cm. This is because after intermixing, at relatively lower electric fields, the electron and hole wave functions are less confined in the QW, resulting in a larger penetration out of the QW. At the relatively higher electric field, the offset of the wave function between the first conduction and the HH valence sub-bands for both the as-grown and intermixed QWs with $L_D$ = 0.45 nm are nearly the same.

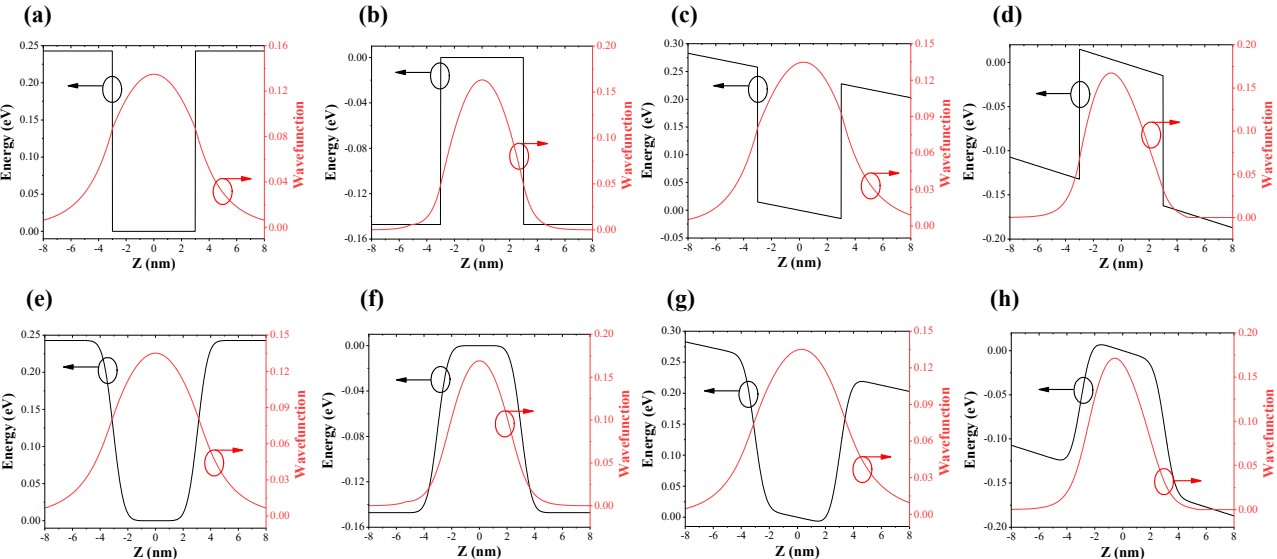

**Figure 3.** Calculated band energy and wavefunction of (**a**–**d**) as-grown QW: (**a**) conduction band at 0 kV/cm electric field, (**b**) HH valence band at 0 kV/cm electric field, (**c**) conduction band at 50 kV/cm electric field and (**d**) HH valence band at 50 kV/cm electric field; (**e**–**h**) for intermixed QW with $L_D$ = 0.45 nm, (**e**) conduction band at 0 kV/cm electric field, (**f**) HH valence band at 0 kV/cm electric field, (**g**) conduction band at 50 kV/cm electric field and (**h**) HH valence band at 50 kV/cm electric field.

The *ER* of the *EAM* can be calculated from:

$$ER = 10log_{10}(\exp(_c\alpha(\omega, F)L_{EAM})) \qquad (14)$$

where $\Gamma_c$ is the optical confinement factor which is calculated to be 5% [32]. Depending on the calculated absorption spectrum and biased voltage data in Figure 2b, the ERs for as-grown and intermixed QWs are obtained and shown in Figure 5. Figure 5a compares the simulated ER for the as-grown and intermixed QWs at 1.55 μm wavelength. The calculated 7.6 dB insertion loss at 0 V bias for the as-grown EAM is close to the measured result of 7.97 dB reported in [33] by using the Hakki–Paoli technique, where the measured net modal absorption coefficient at 1.55 μm wavelength was around 120/cm. There is a dramatic reduction in the insertion loss at 0 V bias for the intermixed QW structures compared with the as-grown QW, which is reduced from 7.6 dB to 0.11dB. The ER reaches −40 dB at −2.4 V bias with $L_D$ = 0.45 nm for 1.55 μm wavelength. If $L_D$ is increased from 0.45 nm to 0.62 nm, i.e., the QW has a 20 nm blueshift, then the maximum ER will be further reduced, and the corresponding reverse voltage will be increased from −2.4 V to −2.9 V. Figure 5b shows that the maximum ER is reduced to 27 dB and 20 dB, respectively, for $L_D$ = 0.45 nm QW when the length of the EAM is reduced from 150 μm to 100 μm, and 70 μm, respectively.

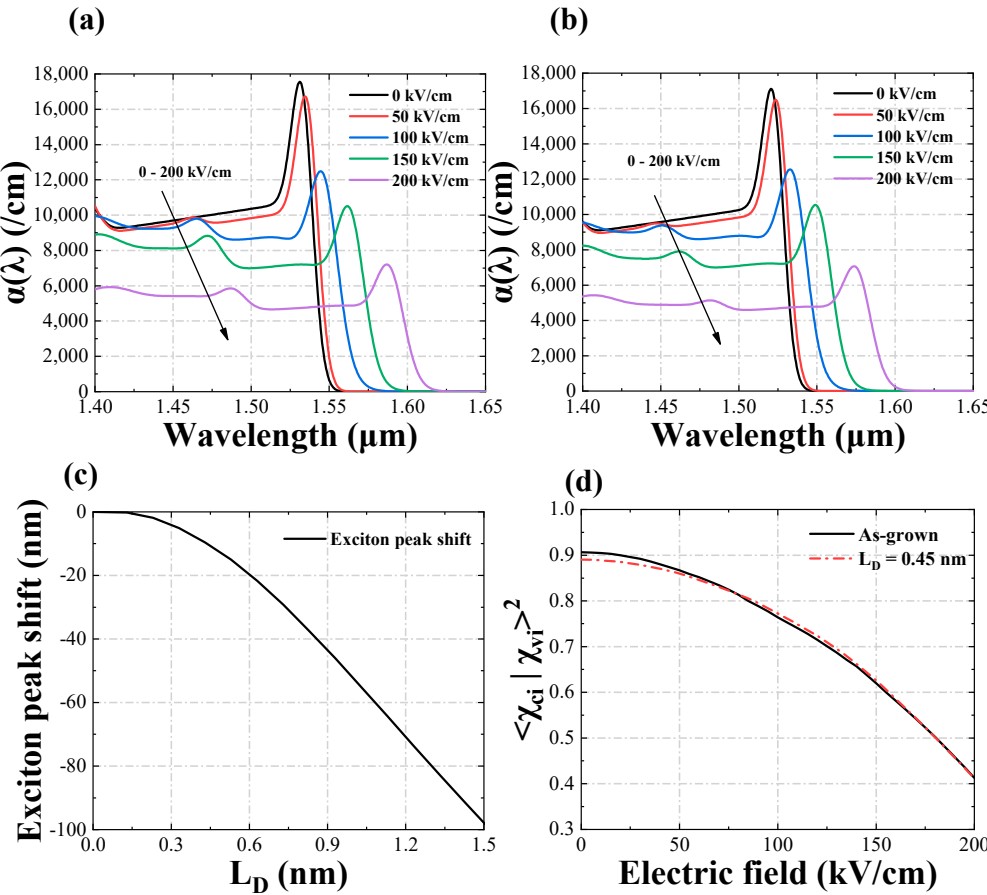

**Figure 4.** TE polarized absorption coefficient spectrum for (**a**) as-grown QW, (**b**) QW with diffusion length $L_D$ = 0.45 nm, (**c**) exciton peak wavelength shift as a function of $L_D$ and (**d**) square of the overlap integral of the wavefunctions between the first conduction and the HH valence sub-bands as a function of electric field.

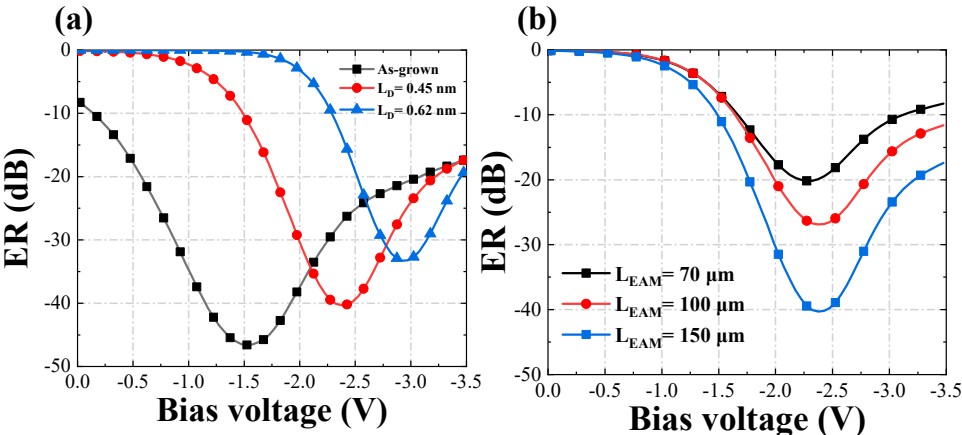

**Figure 5.** (**a**) Calculated ERs for as-grown and intermixed QWs at 1.55 μm wavelength and (**b**) calculated ERs for $L_D$ = 0.45 nm intermixed QW at 1.55 μm wavelength for different EAM lengths.

Figure 6 presents the calculated chirp factor ($\alpha_H$) for as-grown and $L_D$ = 0.45 nm QW. For the as-grown QW, $\alpha_H$ tends to have negative infinity at 1.531 μm and then moves in a positive direction with increasing wavelength for 0 kV/cm electric field, and the zero point of $\alpha_H$ moves to longer wavelengths as the electric field increases, as shown in Figure 6a. In Figure 6b, the dependence of $\alpha_H$ on the electric field at 1.55 μm and 1.56 μm wavelength is indicated, and $\alpha_H$ reduces to 0 at 60 kV/cm. For the intermixed QW with $L_D$ = 0.45 nm, it is

found that the wavelength corresponding to infinitely large $\alpha_H$ is blue shifted by 10 nm to 1.521 μm as in Figure 6c, which causes $a_H$ to increase to more than 4 for 1.55 μm at 0 kV/cm as shown in Figure 6d. It is worth noting that there is no deterioration of the $\alpha_H$ value due to the QWI.

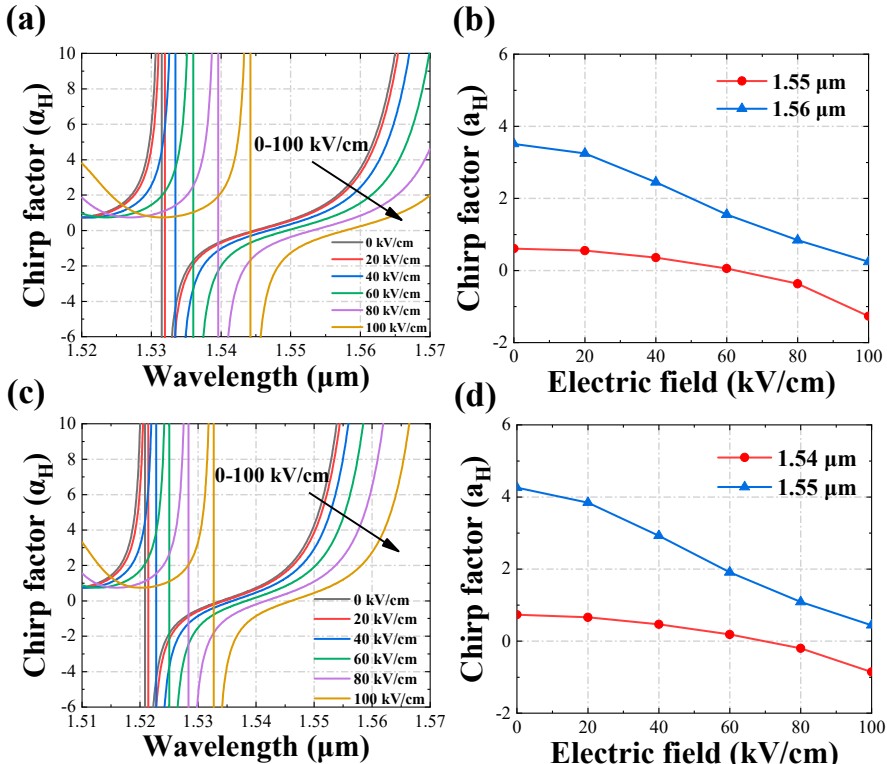

**Figure 6.** (**a**) Simulated chirp factor of as-grown QW as a function of wavelength and (**b**) chirp factor for as-grown QW as a function of the electric field for operating wavelengths of 1.55 μm and 1.56 μm; (**c**) simulated chirp factor of intermixed QW with $L_D$ = 0.45 nm as a function of wavelength and (**d**) chirp factor of intermixed QW as a function of electric field for operating wavelengths of 1.54 μm and 1.55 μm.

To simulate the frequency response of the EAM with QWI, an equivalent circuit [34], shown in Figure 7a, was developed using ADS software. $L_o$ is the inductance of the connecting terminal microstrip; $R_o$ is the load resistance; $C_0$ is the capacitance of the submount; and $R_s$ is the contact resistance which can be measured from the forward current-voltage curve. $R_j$ is the leakage resistance and $C_j$ is the junction capacitance. $C_p$ is the parasitic capacitance of the p-contact, which can be reduced by passivation with hydrogen silsesquioxane (HSQ) [35]. The values of these parameters were measured for as-grown EAM and are: $C_0$ = 0.09 pF, $R_j$ = 10 kΩ, $C_j$ = 0.17 pF, $R_s$ = 24 Ω, $C_p$ = 0.14 pF, $L_o$ = 0.4 nH and $R_o$ = 50 Ω at $V_b$ = −1.6 V. Although the measured result is for a non-QWI structure, the frequency response of a QWI device would be very similar because intermixing has little effect on the dielectric constant and the cut off frequency is not dependent on the precise value of $R_j$. Figure 7b shows the value of $C_j$ and the −3 dB bandwidth as a function of $V_b$. When bias voltage is increased, $C_j$ is reduced because the space charge region easily penetrates the undoped layer due to the high doping (~1×10$^{18}$ cm$^{-3}$) in the P- and N-cladding layers and the relatively thin undoped layer (210 nm), and $C_p$ is kept constant at 0.14 pF. Therefore, the total capacitance is decreased and the −3 dB bandwidth is increased. Figure 7c presents the −3 dB bandwidth and maximum ER as a function of EAM length ($L_{EAM}$), with $V_b$ at −2.4 V and waveguide width ($W_{EAM}$) at 2.5 μm. When the EAM length is reduced, the −3 dB E/O response bandwidth increases at the expense of maximum ER for relatively low reverse EAM voltages. Figure 7d presents the simulated electrical to optical

(E/O) frequency response for the 150 µm-long EAM with $L_D$ = 0.45 nm and shows a −3 dB bandwidth of 22 GHz at $V_b$ = −1.6 V. This result is close to the measured −3 dB bandwidth of 19 GHz reported for an as-grown EAM [7]. The results also show that if the length of EAM is reduced to 70 µm and the ridge waveguide width is reduced from 2.5 µm to 2.0 µm, the −3 dB bandwidth will increase to 41 GHz due to the reduced junction capacitance. The DFB laser ridge waveguide width has to be reduced to 2 µm as well and the grating recess depth needs to be optimized to increase the grating coupling coefficient while maintaining a stable transverse mode.

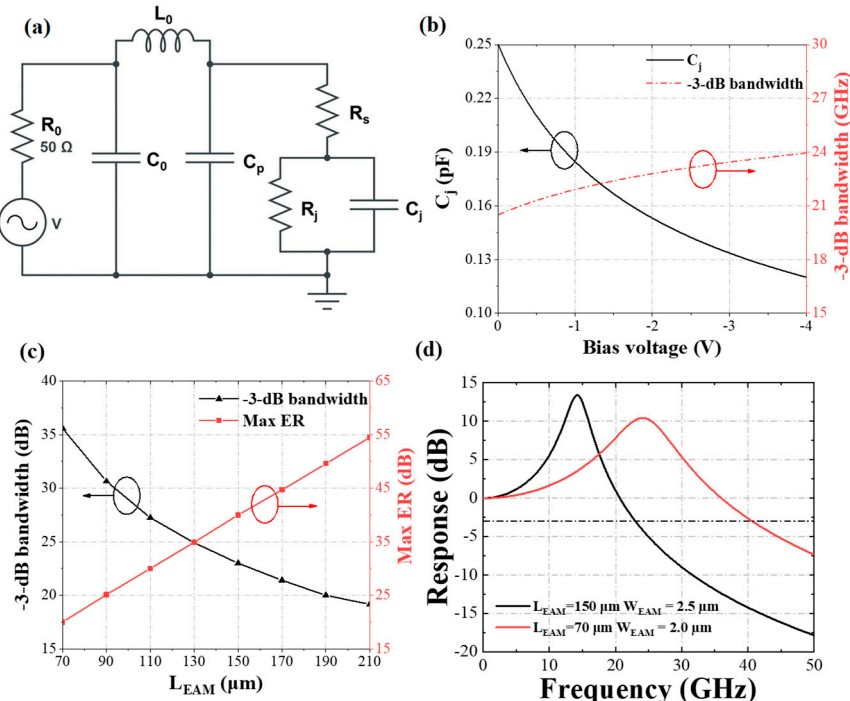

**Figure 7.** (**a**) Equivalent circuit of the EAM; (**b**) $C_j$ and −3 dB bandwidth as a function of bias voltage ($V_b$); (**c**) −3 dB bandwidth and maximum ER as a function of EAM length ($L_{EAM}$) with $V_b$ = −2.4 V and $W_{EAM}$ = 2.5 µm; (**d**) simulated frequency response of 70 µm- and 150 µm-long EAMs.

## 4. Conclusions

A novel design of an AlGaInAs/InP electro-absorption modulator has been proposed. QWI is used in the EAM to enlarge the bandgap to minimize the insertion loss at zero bias. The extent of the QWI process is characterized by the diffusion length of the QW. The absorption spectra and ER for the as-grown and two intermixed QW structures were investigated. The results showed that for a 150 µm-long EAM with a 10 nm blueshift, the ER is 40 dB at 2.5 V EAM reverse bias for 1.55 µm wavelength and the insertion loss of 0.11 dB for 0 V bias, with no penalty in chirp compared to an as-grown QW. The −3 dB bandwidth was predicted using an equivalent circuit based on the measured parameters of an as-grown EAM. The −3 dB bandwidth of the E/O response was 22 GHz, close to the measured bandwidth of an as-grown EAM. When the EAM length and width are reduced to 70 µm and 2.0 µm, respectively, the –3 dB bandwidth of E/O response can be increased to 41 GHz.

**Author Contributions:** Conceptualization, Y.H., R.Z., S.L., J.X., X.L., J.H.M. and L.H.; Data curation, X.S., A.A.-M. and B.Q.; Investigation, X.S., S.Y. and Y.S.; Software, X.S. and W.C.; Writing—original draft preparation, X.S.; Writing—review and editing, J.H.M. and L.H. All authors have read and agreed to the published version of the manuscript.

**Funding:** This work was supported by the U.K. Engineering and Physical Sciences Research Council (EP/R042578/1) and the Chinese Ministry of Education collaborative project (B17023).

**Institutional Review Board Statement:** Not applicable.

**Informed Consent Statement:** Not applicable.

**Data Availability Statement:** Data available on request due to restrictions e.g., privacy or ethical. The data presented in this study are available on request from the corresponding author. The data are not publicly available as they also form part of an ongoing study.

**Acknowledgments:** We would like to acknowledge the staff of the James Watt Nanofabrication Centre at the University of Glasgow for help in fabricating the devices.

**Conflicts of Interest:** The authors declare no conflict of interest.

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
