# Peer review of "Simulation of an AlGaInAs/InP Electro-Absorption Modulator Monolithically Integrated with Sidewall Grating Distributed Feedback Laser by Quantum Well Intermixing"

_photonics, doi:10.3390/photonics9080564_

Round 1

Reviewer 1 Report

The design method is well described, and the performance and dimensions of the design are in agreement with the state-of-the-art. The introduction is well written, and the design part is clear. The scope of the paper is well aligned with the journal's scope. I strongly recommend it for publication once the questions below are addressed.

1. Why are you using an Al composition around 0.42 to 0,34?

2. can you calculate the capacitance and the bandwidth with bias voltage?

3. How were the wavefunctions in figure 3 where calculated?

4. Typo: put a space after [24] in line 212

5. There is a lack of comparison with the state-of-the-art. E.g., a similar approach using QDs was proposed in 10.1109/LPT.2018.2890641. Can you compare the results and the approaches? The QCSE was measured in QDs in 10.1364/OE.455491. Can you comment on how the QCSE performs in QWs and QDs for this application? Can you comment on operating at high temperatures and over Si? What about the wavelength shift of 10 nm at different temperatures? Why are you using that laser and modulator and not others like described in 10.1109/JQE.2021.3087327, 10.1117/12.513627, 10.1143/JJAP.24.L442?

6. Did you use any commercial simulators? It seems that the reference may be missing.

Reviewer 2 Report

Within the paper Simulation of an AlGaInAs/InP electro-absorption modulator monolithically integrated with sidewall grating distributed feedback laser by quantum well intermixing authors presented interesting results of quantum well intermixing induced by diffusion on properties of novel optoelectronic device. The work is interested and well organized, however during my review I noted some issues that in my opinion have to be addressed before the publication:

- first of all, why authors calculate interdiffusion of Ga and Al instead of In, which is known to diffuse in all Indium containing III-V semiconductor materials?

- secondly, why and how authors decide to use the value of LD equal to 0.45 nm? Why this value is the same for both Ga and Al (both have different diffusion properties).

- the question is how to technologically control the LD? (concerning the Figure 4c)

- I would recommend to add band model graph for better understanding the analysed MQW structure

- Figure 3 - please calculate and add in the figures the overlaps of electrons and holes wavefunction

- I would recommend to use cm-1 instead of /cm

- Figure 4a and 4b are to small to recognize absorption for 1550 nm

- Figure 4c, is the results shown in the graph calculated for nonpolarized device ( 0 kV/cm)?

- table 1 - please add information that last 3 parameters are Luttinger parameters

- in the equation 2 there is no HH and LH so the line 131 is quite confusing, moreover why authors used "r" for description of conduction band?

- lines 157 and 250 should be comma instead of dot

- lines 171, 212 missing spaces

- please provide some details of DFB gratings applied in the device

Author Response

Please see the attachment for reviewer 2

Reviewer 3 Report

In this paper, authors propose an EAM monolithically integrated with a side wall DFB, and simulate its performance for different QWI diffusion length. However, I cannot consider it to be published before the authors well address my concerns.

1.     In your design, DFB is shallow etched and EAM is deep etched. How much is the reflectivity at the interface? Even small reflection will cause the linewidth broadening in DFB especially for QWS structure.

2.     Can author provide more details in QWI technique and what’s the tolerance? Is the transition length short enough for a 30um isolation region?

3.     In Fig.3 g and h, why electron and hole wavefunction shift in the same direction? For the as-grown, both move in the opposite way is reasonable.

4.     What’s the intrinsic bandwidth of EAM? In the calculation of frequency response, both parasitic and intrinsic frequency response are required to be considered.

5.     It would be better to plot 3dB bandwidth vs L and Max ER (within a Vb value) vs L so that the tradeoff between ER and bandwidth is clear.

Author Response

Please see the attachment for reviewer 1

Round 2

Reviewer 2 Report

I am satisfied with author corrections.

Author Response

Thank you for your comments. 

Reviewer 3 Report

Thanks for the response. However, I cannot recommend it to be published in the present status, since two of my concerns are not addressed appropriately.

1.    Authors claim that 20dB back reflection is negligible for linewidth is inappropriate. There are lots of papers talking about back reflection and the coherence of DFB. And there are some designs like partial grating DFB that can improve the feedback tolerance.

2.    For the intrinsic bandwidth, it is the response time caused by the carrier relaxation, etc in the junction instead of the circuit part. It can be greater than the parasitic bandwidth, but authors need to prove that in the analysis.
